# Signals of Holocene climate transition amplified by anthropogenic land-use changes in the Westerly-Indian Monsoon realm

Nicole Burdanowitz[1], Tim Rixen[1,2], Birgit Gaye[1], Kay-Christian Emeis[1,3]

[1]Institute for Geology, Universität Hamburg, Bundesstraße 55, 20146 Hamburg, Germany
[2]Leibniz-Zentrum für Marine Tropenforschung (ZMT), Fahrenheitstraße 6, 28359 Bremen, Germany
[3]Institute of Coastal Research, Helmholtz Center Geesthacht, Max-Planck-Straße 1, 21502 Geesthacht, Germany

*Correspondence to*: Nicole Burdanowitz (nicole.burdanowitz@uni-hamburg.de)

**Abstract.** The Indian Summer Monsoon (ISM) with its rainfall is the lifeline for people living on the Indian subcontinent today and possibly was the driver of the rise and fall of early agricultural societies in the past. Intensity and position of the ISM have shifted in response to orbitally forced thermal land-ocean contrasts. At the northwestern monsoon margins, interactions between the subtropical westerly jet (STWJ) and the ISM constitute a tipping element in the Earth's climate system, because their non-linear interaction may be a first-order influence on rainfall. We reconstructed marine sea surface temperature (SST), supply of terrestrial material and vegetation changes from a very well-dated sediment core from the northern Arabian Sea to reconstruct the STWJ-ISM interaction. The Holocene record (from 11,000 years) shows a distinct, but gradual, southward displacement of the ISM in the Early to Mid-Holocene, increasingly punctuated by phases of intensified STWJ events that are coeval with interruptions of North Atlantic overturning circulation (Bond events). Effects of the non-linear interactions culminate between 4.6 - 3 ka BP, marking a climatic transition period during which the ISM shifted southwards and the influence of SWTJ became prominent. The lithogenic matter input shows an up to 4-fold increase after this time period probably related to the strengthened influence of agricultural activities of the Indus civilization with enhanced erosion of soils. This anthropogenic land-use change is amplifying the impact of Bond events and adding to the marine sedimentation rates adjacent to the continent.

## 1 Introduction

Changes in latitude and strength of the ISM were gradual over the course of the Holocene, but were punctuated by distinct climatic events (Herzschuh, 2006). The ISM was strong during the Early Holocene until approximately 6.5 ka ago, and influenced regions farther north than today (Herzschuh, 2006; Prasad and Enzel, 2006). A weakening of the ISM is reflected in many records after the Holocene Optimum Period, commonly attributed to declining summer insolation (Banerji et al., 2020; Herzschuh, 2006), latitudinal insolation gradients (Mohtadi et al., 2016; Ramisch et al., 2016) and feedbacks of climate anomalies such as the Arctic Oscillation (Zhang et al., 2018), the North Atlantic Oscillation (NAO) (Band et al., 2018; Banerji

et al., 2020; Kotlia et al., 2015; Lauterbach et al., 2014) or the El Niño Southern Oscillation (ENSO) (Banerji et al., 2020; Prasad et al., 2014; Srivastava et al., 2017).

Changes of the ISM system not only affected hydrological conditions and vegetation in this region, but were also an important constraint on the expansion and dispersal of the Indus civilization, also known as Harappan civilization, emerging around 5 ka ago (Durcan et al., 2019; Giosan et al., 2012, 2018; Possehl, 1997). Growing agricultural activities adjacent to the northwestern Arabian Sea increased soil erosion and, therefore, terrigenous sediment input (Gourlan et al., 2020). Further, the decrease of ISM precipitation favored the development of combined summer and winter cropping as well as drought-tolerant crops (Petrie and Bates, 2017). Winter precipitation intensity and interannual variability in the present-day northwestern region of the Indian Monsoon (IM) domain is governed by STWJ-induced events of strong precipitation, so called Western Disturbances (WD) (Anoop et al., 2013; Dimri et al., 2016; Hunt et al., 2018; Leipe et al., 2014; Munz et al., 2017; Zhang et al., 2018). A tracking algorithm of ERA-interim data found about 3000 WDs (6 – 7 per month) between 1979 and 2016 in the region of Pakistan and India. About five percent of the WD had precipitation rates of about 40 mm/day and the rest had about 3 mm/day (Hunt et al., 2018). Further, the frequency of WDs in Pakistan and India during the winter months was higher when the position of the STWJ was shifted to the south (Hunt et al., 2018). Therefore, a latitudinal change in the position of the STWJ during the past should have had a strong impact on the frequency of WDs and the amount of winter precipitation over that region. Climatological data show a link between the NAO, the position of the STWJ and the strength of the WD during the winter over northwest India (Yadav et al., 2009). A positive NAO is associated with an intensified STWJ and stronger WD and, thus, above normal winter precipitation in this region (Yadav et al., 2009). The interaction, position and strength of the IM and the STWJ on millennial to Holocene timescales are yet poorly understood, because the pronounced signal of the ISM dominates sedimentary records, and high-quality records in the region where IM and STWJ interact are sparse.

The northeastern (NE) Arabian Sea (AS) is located in the region of interaction of IM and STWJ. Whereas the primary productivity during the warm ISM season is driven by lateral transport of nutrient-rich waters from the Oman upwelling area (Schulz et al., 1996), the NE winds deepen the mixed layer during the colder IWM season due to convective mixing leading to enhanced primary productivity and carbon exports to the deep sea (Banse and McClain, 1986; Madhupratap et al., 1996; Rixen et al., 2019). The NE AS holds high-quality archives due to excellently preserved and highly resolved sediment records deposited on the shelf and slope impinged by a strong oxygen minimum zone (OMZ) between 200 m and 1200 m water depth (von Rad et al., 1995; Schott et al., 1970; Schulz et al., 1996). Most existing records have poor temporal resolution or do not bracket the entire Holocene (Böll et al., 2014, 2015; Giosan et al., 2018; Lückge et al., 2001; Munz et al., 2015), but they do outline the interplay of the Indian Winter Monsoon (IWM) with the westerlies in the NE AS realm during the Holocene and suggest that winter monsoon activity intensified between 3.9 and 3.0 ka BP (Giesche et al., 2019; Giosan et al., 2018; Lückge et al., 2001) and intermittently during the last ca. 1.5 ka (Böll et al., 2014; Lückge et al., 2001; Munz et al., 2015).

For a detailed and high-resolution reconstruction, we have analyzed lithogenic mass accumulation rates (LIT MAR), alkenone-based SST and *n*-alkane-based land vegetation from the NE AS (Figure 1) over the last 10.7 ka. They document the influence

of the IWM and the westerlies on the marine and terrestrial environment in the NE AS region and show that the advent of agriculture in step with climate change significantly enhanced soil erosion.

65

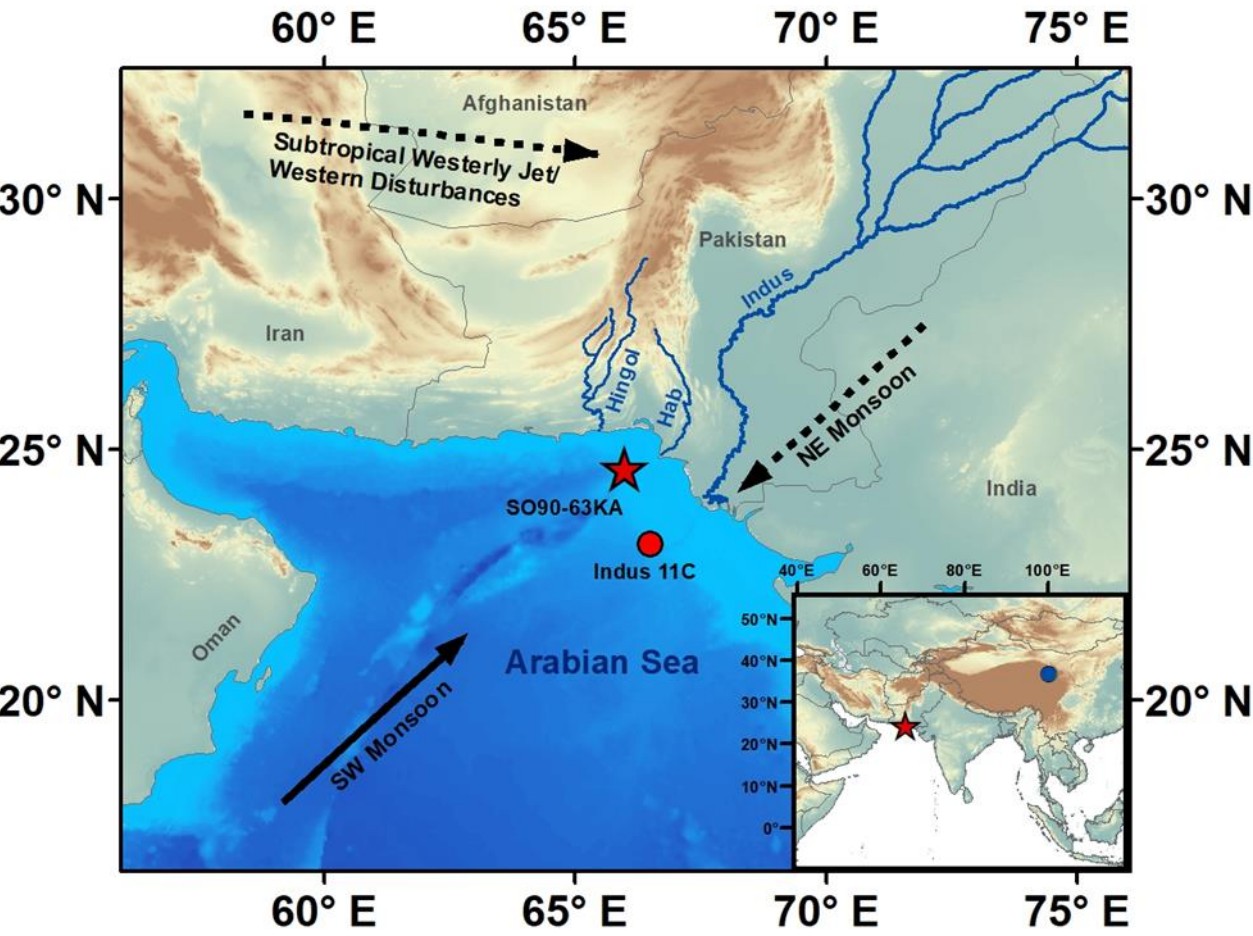

Figure 1: Location of the study site SO90-63KA (red star), the nearby record Indus 11C (Giosan et al., 2018) (red circle) and Lake Qinghai (Hou et al., 2016) (blue dot in the inset map). The rivers (blue lines) important for the study site are shown. Prevailing wind pattern during the summer monsoon, here indicated as southwest (SW) monsoon, is shown with a black arrow. The prevailing wind pattern of the winter monsoon, here indicated as northeast (NE) monsoon as well as the subtropical westerly jet and its associated rain bearing Western Disturbances, are shown as black dashed arrows. The map was created using ArcGIS v.10.8 (ESRI, 2019). The bathymetric data are from the General Bathymetric Chart of the Oceans (GEBCO2014; www.gebco.net).

## 2 Material and Methods

Records presented here were raised from the box core SO90-63KA (697 cm long), which was retrieved from the Arabian Sea off Pakistan (24°36.6'N, 65°59.0'E, 315 m water depth) during the RV SONNE cruise SO90 in 1993. With the upper 18 cm missing, SO90-63KA covers the last ca. 10.7 ka and encompasses the development of early Bronze Age agricultural societies (Staubwasser et al., 2002, 2003). The chronology is based on 80 $^{14}$C dates of planktic foraminifers *Globigerinoides sacculifer* (Fig. S1) (Staubwasser et al., 2002, 2003) and belongs to the best dated cores from the Arabian Sea. The major and trace elements of SO90-63KA were published previously (Burdanowitz et al., 2019; Staubwasser and Sirocko, 2001).

### 2.1 Lipid analyses (alkenones and *n*-alkanes)

For the lipid analyses, 2.0 to 13.4 g of freeze-dried and ground sediment were extracted with a DIONEX Accelerated Solvent Extractor (ASE 200) at 75°C, 1000 psi for five minutes using dichloromethane (DCM) as solvent. The procedure was repeated three times. A known amount of an internal standard (14-heptacosanone, squalane) was added prior to extraction. An additional working sediment standard was extracted for each ASE running sequence. The total lipid extracts (TLEs) were concentrated using rotary evaporation. The TLEs were separated into a hexane-soluble and a hexane-insoluble fraction by $Na_2SO_4$ column chromatography. The hexane-soluble fractions were saponified at 85°C for two hours using a solution of five percent potassium hydroxide in methanol (MeOH). Afterwards, the neutral fraction was extracted with hexane. Column chromatography using deactivated silica gel (five percent $H_2O$, 60 mesh) was carried out to separate the neutral fraction into an apolar- (containing *n*-alkanes), ketone- (containing alkenones) and polar-fraction by using hexane, DCM and DCM:MeOH (1:1), respectively. The apolar fraction was further cleaned by $AgNO_3$-Si column chromatography using hexane as solvent.

Quantifications of the *n*-alkanes and the alkenones were carried out using an Agilent 6850 gas chromatograph (GC) equipped with an Optima1 column (30 m, 0.32 mm, 0.1 µm), split/splitless injector operating at 280°C and a flame ionization detector (FID, 310°C). $H_2$ was used as carrier gas with a flow rate of 1.5 ml/min. Samples were injected in hexane and duplicate measurements of each sample were carried out for the alkenones. For the *n*-alkanes, the GC temperature was programmed from 50°C (held 1 min) ramped at 8°C/min to 320°C (held 15 min). An external standard was used for quantification, containing *n*-$C_8$ to *n*-$C_{40}$ alkanes in known concentration. Repeated analyses of the external standard resulted in a quantification precision of 6 % (1 SD).

The average chain length (ACL) of the homologues $C_{27}$-$C_{33}$ was calculated using following equation:

$$ACL_{27-33} = \frac{27 * C_{27} + 29 * C_{29} + 31 * C_{31} + 33 * C_{33}}{C_{27} + C_{29} + C_{31} + C_{33}}$$

where $C_x$ is the concentration of the *n*-alkane with x carbon atoms.

The carbon preference index (CPI) of the *n*-alkanes was calculated as:

$$CPI_{27-33} = 0.5 * \left( \frac{C_{27} + C_{29} + C_{31} + C_{33}}{C_{26} + C_{28} + C_{30} + C_{32}} + \frac{C_{27} + C_{29} + C_{31} + C_{33}}{C_{28} + C_{30} + C_{32} + C_{34}} \right)$$

where $C_x$ is the concentration of the *n*-alkanes with x carbon atoms.

For the alkenones, the GC temperature was programmed to increase from 50°C (held 1 min) to 230°C at 20°C/min, then at 4.5°C/min to 260°C and at 1.5°C/min to 320°C (held 15 min). The $C_{37:2}$ and $C_{37:3}$ alkenones were identified by comparing the retention time peaks of the samples and the known working sediment standard. Quantification was carried out by using a known amount of an external standard (14-heptacosanone and hexatriacontane).

We have calculated the alkenone unsaturation index using the following equation (Prahl et al., 1988):

$$U_{37}^{k\prime} = \frac{C_{37:2}}{C_{37:2} + C_{37:3}}$$

and using the core top calibration of Indian Ocean sediments (Sonzogni et al., 1997) to convert the UK'37 index to SSTs:

$$SST = \frac{U_{37}^{k\prime} - 0.043}{0.033}$$

For each sample, at least one duplicate measurement was obtained with an average precision of 0.1°C (1 SD). The precision of the replicate extractions of the working standard sediment (n=13) and duplicate measurements of each replicate was 0.5°C (1 SD).

## 2.2 Lithogenic mass accumulation rates

The total mass accumulation rates (MAR) were calculated by multiplying the dry bulk density (DBD) with the linear sedimentation rate (LSR). The LSR was derived from depth increments of successive age model tie points. DBD was calculated with the following equation (Avnimelech et al., 2001):

$$DBD = \frac{weight\ dry\ sample\ (Wd)}{total\ sample\ volume\ (Vt)}$$

where

$$Vt = volume\ of\ solids + volume\ of\ water$$

$$= \frac{weight\ dry\ sample}{particle\ density} + (weight\ wet\ sample - weight\ dry\ sample)$$

We have assumed 2.65 g/cm³ as the density of inorganic sediment particles (Blake and Hartge, 1986; Boyd, 1995) and 1 g/cm³ for water density, and corrected for organic matter contents assuming a density of 1.25 g/cm³ for organic particles (Boyd, 1995):

$$sediment\ particle\ density_{weighted\ average} = 1.25 * \%OM + 2.65 * (100 - \%OM)$$

It has to be noted that we did not calculate the LIT MAR between 9.4 and 8.5 ka BP because the core was dried out in this section.

The wavelet power spectrum analysis of the LIT MAR record was carried out using a Matlab Wavelet script (Torrence and Compo, 1998). As uniformly distributed time steps are necessary to perform the analysis we have interpolated our data set to achieve a temporal resolution of about 15.6 years. This is the minimum time step between two samples and also covers the

gap between 9.4 and 8.5 ka BP. We are aware that the uncertainty may be high, especially during the gap period where no data for LIT MAR exist.

### 2.3 Grain size analyses

A coarser resolution data set of grain size analyses of SO90-63KA was previously published (Burdanowitz et al., 2019). Here, we have used an extended data set of 146 sediment samples for the grain size analyses of SO90-63KA and for the endmember modelling analyses. The grain size analyses were carried out at the Center for Tropical Marine Research (ZMT, Bremen). To remove inorganic carbon and organic matter 250 mg of bulk sediment were treated with muriatic acid (HCl) and hydrogen peroxide ($H_2O_2$) (e.g. Sun et al., 2002). Afterwards, the samples were treated with Calgon ® ($Na_6P_6O_{18}$) and boiled to avoid particle coagulation. The grain sizes were then measured with a Horiba LA-950V2 laser scattering particle size analyser. A triplicate measurement of each sample was carried out and results were averaged. The grain size distribution was calculated for 93 grain size classes between 10 nm and 3 mm and grain size distribution parameters were calculated with GRADISTAT V.8. The end-member modelling analysis (EMMA) Matlab script by Dietze et al. (2012) was used for the end-member modelling.

## 3 Results

### 3.1 Alkenones and SST record of SO90-63KA

SST estimates based on the UK'37 index reconstructed SST vary by about 2°C from 25.9°C to 27.9°C (UK'37: 0.896 to 0.963) throughout the Holocene at the core site (Figure 2). Periods of low SSTs were the Early Holocene until ca. 8.7 ka BP and the time interval between ca. 4.6 and 3.0 ka. A marked decrease of about 1°C occurred between 4.2 and 3.8 ka BP. High SSTs were registered between 8.1 and 5.2 ka BP during the Mid-Holocene. The last 3 ka show also high and variable SSTs.

### 3.2 *n*-Alkane record of SO90-63KA

The *n*-alkane patterns of sediment samples (n=93) display a strong odd-over-even carbon number predominance with $CPI_{27-33}$ values of 2.1 to 7.6 (Fig. S2). The *n*-alkane $C_{31}$ is the most abundant component (mean: $25.5 \pm 2.1$ %) followed by *n*-$C_{29}$ (mean: $17.0 \pm 1.8$ %) and *n*-$C_{33}$ (mean: $12.2 \pm 2.3$ %) for most samples (Fig. S3). The $ACL_{27-33}$ ranges between 29.9 and 30.7 with highest values between 9.2 and 6.8 ka BP and lowest values from about 6.8 to 3.6 ka BP during the Mid-Holocene (Fig. S2). The $ACL_{27-33}$ values increase slightly during the last about 3.5 ka, but on average, do not reach the high values characteristic for the Early Holocene period.

### 3.3 Lithogenic mass accumulation rates (LIT MAR) of SO90-63KA

The LIT MAR in our record range between 225 g/(m²*yr) and 3714 g/(m²*yr) (Figure 2). Periods of high LIT MAR occurred from 10.1 – 9.8 ka BP, at ca. 8.3 ka BP, around 3.4 ka BP, between 2.8 - 2.7 ka BP, 2.2 - 2.1 ka BP, 1.5 – 1.1 ka BP and over the last 300 years. The LIT MAR are relatively low in the Mid-Holocene period from 8.3 to 3.5 ka BP with only minor fluctuations compared to the rest of our record. LIT MAR are mostly controlled by the sedimentation rate of SO90-63KA and less by the dry bulk density and the content of lithogenic material (Fig. S1), although the lithogenic material content slightly increases from the Mid- to Late Holocene.

### 3.4 Grain size and end-member modelling analyses of SO90-63KA

The grain size analyses of SO90-63KA (n=146) show a bimodal pattern with modal grain sizes of 7.2 µm and 0.5 µm (Fig. S4). The median grain sizes of all samples range between 4.6 µm and 10.0 µm (mean: 6.3 ± 1.0 µm) and the EMMA segregates three main end-members (Fig. S5). End-member 1 (EM1) presents a polymodal pattern with modal grain sizes of 5.5 µm, 24.5 µm and 0.4 µm. End-member 2 (EM2) has a bimodal pattern with modal grain sizes of 16.27 µm and 0.4 µm, whereas end-member 3 (EM3) has a trimodal pattern with modal grain sizes of 0.4 µm, 4.2 µm and 42.1 µm (Fig. S5).

## 4 Discussion

### 4.1 Holocene climate change in the NE Arabian Sea realm

The prime indicator for climate- and human-driven change on land adjacent to the NE AS is the lithogenic material archived in the sediment record. The lithogenic material at the core site is supplied by fluvial input from the Makran rivers (Forke et al., 2019; Lückge et al., 2002; von Rad et al., 2002; Staubwasser and Sirocko, 2001; Stow et al., 2002) and by aeolian input that originates from the Sistan Basin region in the border region of Afghanistan and Iran (Burdanowitz et al., 2019; Forke et al., 2019; Kaskaoutis et al., 2014; Rashki et al., 2012). At present, dust from the Sistan region is transported to the NE AS by northerly Levar winds during summer and northeasterly and westerly winds during winter (Hussain et al., 2005; Kaskaoutis et al., 2015; Pease et al., 1998; Rashki et al., 2012; Sirocko et al., 1991; Tindale and Pease, 1999). Ti/Al ratios of SO90-63KA are an indicator for fluvial input by the Makran rivers (Burdanowitz et al., 2019) and helps to distinguish between aeolian and fluvial input. Ti along the Makran margin is enriched in terrestrial detritus mainly delivered from rivers (Lückge et al., 2001) and Ti/Al ratios are particularly high in fluvial deposited sediments along the Hingol River. In thick varves Ti/Al ratios are enriched compared to the thin varves indicating enhanced transport energy as a result of a more intense fluvial discharge (Lückge et al., 2001). Another useful proxy is end-member 3 (EM3) of the EMMA as an indicator for aeolian input (Burdanowitz et al., 2019). Dust storms over Karachi, located east of the Hab River mouth, exhibit median grain sizes ranging between 2.4 and 3.4 µm (Fig. S4) and EM3 have modal grain sizes of 4.2 µm (Fig. S5, this study) and 3.7 µm (Burdanowitz et al., 2019), respectively. The modal grain size of 4.2 µm is similar to that of the dust over Karachi and can thus be used as

an indicator for aeolian input. Alkenone based SSTs reconstructed from our record range between 25.9 and 27.9°C throughout

the Holocene. It has been pointed out that changes in the alkenone producing coccolithophore communities *Emiliania huxleyi* and *Gephyrocapsa oceanica* in the NE AS are controlled by nutrient availability and mean mixed layer depth (Andruleit et al., 2004). This variation in the coccolithophore communities over the course of the year results in higher alkenone fluxes during winter and spring compared to other seasons. However, investigations of seasonal data on alkenone temperatures (Böll et al., 2014) showed that the alkenone-based SST reconstructions from sediments are representative for annual mean SST in our

study area. Our Holocene alkenone based SST record shows a similar pattern as the nearby planktic foraminifera DNA based SST/winter monsoon record of core Indus 11C near the Indus River mouth (Giosan et al., 2018) (Figure 2b).

During waning glacial conditions of the Early Holocene the SSTs remained low until ca. 8.7 ka BP in the NE AS (Böll et al., 2015; Gaye et al., 2018; Giosan et al., 2018) (Figure 2). Between 8.7 and 8.1 ka BP SST increased from 26.0°C to 27.2°C with rising LIT MAR around 8.4 to 8.3 ka BP. However, since there is a gap in our LIT MAR record, as mentioned before, we

cannot exclude an onset of rising LIT MAR even prior to 8.4 ka BP. Coeval rapid changes have been identified across the entire IM realm (Dixit et al., 2014; Rawat et al., 2015) and mark the so-called 8.2 ka event, which was associated with a weakened ISM (Cheng et al., 2009; Dixit et al., 2014).

The North Atlantic hematite stained grain (HSG) records indicate eight pronounced cold events, so called Bond events, during the Holocene (Bond et al., 2001). Recent studies have suggested that changes in the NAO and the Atlantic Meridional

Overturning Circulation (AMOC) may have played important roles causing at least some of these cold events in the North Atlantic region (Ait Brahim et al., 2019; Goslin et al., 2018; Klus et al., 2018). Whereas some Bond events (0, 1, 5, 7 & 8) occurred during negative NAO-like conditions, linked to a reduction in North Atlantic Deep Water formation, the other Bond events (mainly during the Mid-Holocene) occurred during positive NAO-like conditions (Ait Brahim et al., 2019). The weakened ISM of the 8.2 ka event has been linked to one of these North Atlantic cool phases (Bond event 5) and is interpreted

to signal a decrease in ocean heat transport that induced a marked southward shift of the Intertropical Convergence Zone (ITCZ) (Cheng et al., 2009).

Although our SST record shows no evidence of the 8.2 ka cold event in the NE AS, $\delta^{18}$O data of *Globigerinoides ruber* of the same core interval signal slightly declining Indus river discharge around 8.4 ka BP (Staubwasser et al., 2002) (Figure 2), accompanied by slightly declining Ti/Al ratios from 8.6 to 8.2 ka BP and rising LIT MAR around 8.4 ka BP indicative for

strong soil erosion due to enhanced aridity on land. However, this signal is not very pronounced in the SO90-63KA record and the peak around 8.4 ka BP in the LIT MAR record is mostly controlled by the sedimentation rate (Fig. S1). High SSTs prevailed after the Early-Mid Holocene transition from ca. 8.1 ka BP until ca. 5.2 ka BP and are consistent with other Arabian Sea records (Böll et al., 2015; Gaye et al., 2018; Giosan et al., 2018). The warm period encompasses the Mid-Holocene climate optimum period and is characterized by high precipitation rates in the North Himalayan region (Dallmeyer et al., 2013;

Herzschuh, 2006). High Ti/Al ratios thus reflect increasing fluvial inputs, although low LIT MAR indicate low soil erosion (Figure 2). The decoupling probably was due to a dense vegetation cover during this Mid-Holocene wet phase which also

explains why the LIT MAR record shows no correlation to the Bond events during the Mid-Holocene. The regional vegetation has damped their remote influence.

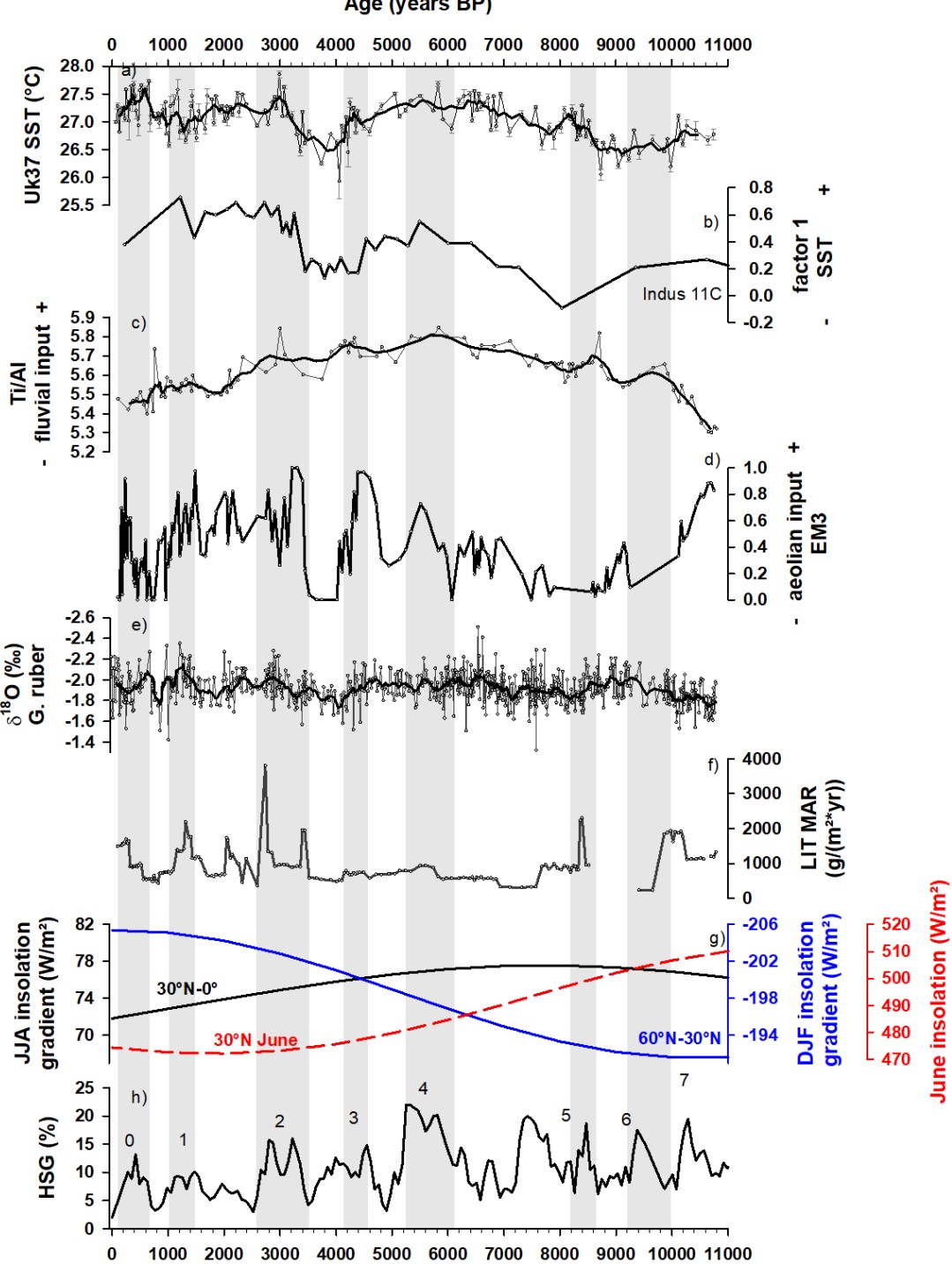

 **Figure 2: Environmental changes in the NE Arabian Sea. a) alkenone-based reconstructed SST of SO90-63KA, b) factor 1 based on DNA factor analyses of planktic foraminifera DNA interpreted as SST/winter monsoon indicator in core Indus 11C (Giosan et al., 2018). c) Ti/Al ratios indicating fluvial sediment input, d) EM3 based on end-member modelling analyses of grain sizes as an indicator for aeolian input (Burdanowitz et al., 2019) and f) lithogenic mass accumulation rates (LIT MAR) of SO90-63KA. e) $\delta^{18}O$ of *Globigerinoides ruber* of SO90-63KA (Staubwasser et al., 2003). g) latitudinal insolation gradients (LIG) of summer (JJA: 30°N – 0°, black line) and winter (DJF: 60°N – 30°N, blue line) and summer insolation at 30°N (dashed red line) (Laskar et al., 2004). h) stacked North Atlantic hematite stained grains (HSG) representing drift-ice in core MC52-V29191 with so called "Bond events" (numbered grey bars) (Bond et al., 2001). Thick black lines in a) and c) indicate five-point running averages.**

From about 4.6 ka BP onwards a cooling period began with a moderate initial cooling event at 4.4 ka BP and a rapid SST decrease of about 1°C between 4.2 and 3.8 ka BP. Foraminiferal analyses of the same core imply decreased Indus river discharge and increased IWM mixing between 4.5 – 4.25 ka BP (Giesche et al., 2019; Staubwasser et al., 2003) which coincided with enhanced eolian input (EM3, Figure 2). Giesche et al. (2019) postulated a weakening of the IWM strength from 4.1 to 3.9 ka BP and a period of cooling between 3.7 and 3.3 ka BP which is indicated by salinity sensitive foraminera. This partly contradicts our SST record. However, the $\delta^{18}O$ record of *G. ruber,* mainly interpreted as a salinity signal, is also affected by water temperature (Giesche et al., 2019; Staubwasser et al., 2003). If the $\delta^{18}O$ signals were interpreted as a temperature rather than a salinity signal, Giesche et al. (2019) argued that the observed increase of the $\delta^{18}O$ around 4.1 ka BP would be consistent with a surface water cooling of about 1°C. This interpretation matches our alkenone-based SST reconstruction at that time. Although considering both variations of the SST and $\delta^{18}O$ records may be partly within the methods uncertainties, we are convinced that they show a climatic signal. They are independent of each other and, in addition, the EM3 record shows a rapid change during that time. Our record shows that low SSTs prevailed until 3.0 ka BP and occurred along with decreased fluvial and aeolian input as shown by the Ti/Al ratios and EM3. The marked cooling between 4.6 and 3.0 ka BP is widespread in the Arabian Sea and on the Indian subcontinent (Gaye et al., 2018; Zhao et al., 2017).

Overall, long-term trends in the Ti/Al ratios and LIT MAR are driven by a strong ISM influence during the Early Holocene and a gradual increase (decrease) of STWJ (ISM) influence from the Mid- to Late Holocene. The aridification trend from the Mid- to Late Holocene has led to a generally decreased fluvial input during the Late Holocene (Burdanowitz et al., 2019). The last ca. 3.0 ka in the record are characterized by high and variable SSTs, highly variable LIT MAR with strong peaks, and increasing aeolian inputs, whereas fluvial inputs of the Makran rivers decrease (Burdanowitz et al., 2019). The highly variable LIT MAR record might be a combination of variable sedimentation rates and dry bulk density. For instance, the high LIT MAR around 3.4 and 2.8 ka BP coincide with high sedimentation rates but not a strong change in the dry bulk density or lithogenic material content (Fig. S1). However, the high LIT MAR around 1.4 ka BP coincide with slightly increased sedimentation rates and increased dry bulk density. In contrast to the generally smoothly decreasing river discharges, the higher SSTs and especially LIT MAR and aeolian inputs, reveal pronounced variations (Figure 2). Furthermore, short episodes of high fluvial sediment discharge interrupt the general trend of decreasing fluvial input. The variations appear to be associated with a 1425 cycle, similar to the Bond events, that is superimposed on this long-term trend (Burdanowitz et al., 2019). We posit, that the combination of pronounced soil erosion, flood events due to precipitation events, and strong winds during arid phases have led to the high LIT MAR in the Late Holocene part of our record. These changes are mainly associated with

decreasing ISM activity as well as increasing wind in the source areas of the aeolian material and variable IWM strength (Böll et al., 2014, 2015; Burdanowitz et al., 2019; Ivory and Lézine, 2009; Lückge et al., 2001). The main forcing mechanisms have been suggested to be the southward shift of the ITCZ due to decreasing summer solar insolation, thermal land-ocean contrast and/or teleconnection to mid-high-latitude NH climate probably via the STWJ (Böll et al., 2014, 2015; Burdanowitz et al., 2019; Fleitmann et al., 2007; Giosan et al., 2018; Lückge et al., 2001; Mohtadi et al., 2016; Munz et al., 2015, 2017). A strong impact of NH climate is indicated by high accumulation rates during Bond events 0-2 (Figure 2): The up to 4-fold increase of LIT MAR during the Late Holocene Bond events marks a system shift towards a strong response to mid-high latitude NH climate via the STWJ increasing precipitation in winter, but decreasing precipitation during the summer. Furthermore, this change in the seasonal precipitation pattern enhanced soil erosion due to stronger erosive forces of rivers in largely derelict farmlands exposed to desertification due to an overall increased aridification trend (Giosan et al., 2012, 2018; Wright et al., 2008).

## 4.2 Transition period from low latitude to increased mid-high latitude influence on the Makran coast

The smooth and long-term decrease of the summer insolation at 30°N after ca. 11 ka BP is an important driver of climate and vegetation changes in the IM and westerlies realm (Fleitmann et al., 2007; Herzschuh, 2006). It does, however, not explain the marked cooling event between about 4.6 and 3 ka BP and the subsequently unstable warm period seen in our record. The smoothly decreasing summer insolation cannot be the sole driver of these delayed and fast-paced environmental changes even if one considers that the maximum ISM intensity lags the maximum summer insolation by about 3 ka (Ansari and Vink, 2007; Fleitmann et al., 2007; Leipe et al., 2014; Reichart et al., 1998; Zhang et al., 2018; Zhao et al., 2017). A decisive influence instead may be the latitudinal insolation gradient (LIG) that triggers changes of the atmospheric pressure gradient, heat transport and determines the strength and position of regional atmospheric circulation (Bosmans et al., 2012; Davis and Brewer, 2009; Lee and Wang, 2014; Mohtadi et al., 2016; Wang et al., 2017). This has so far received little attention in paleostudies (Clemens and Prell, 2003). The cross-equatorial insolation gradients play an important role for inter-hemispheric moisture transport and the extent of the Hadley cell and summer cross-equatorial insolation gradients may drive glacial-interglacial variability (Bosmans et al., 2015; Reichart, 1997).However, Davis and Brewer (2009) argued that intra-hemispheric summer LIG and the latitudinal temperature gradient (LTG) also influence the strength of the monsoon system and its poleward displacement and, respectively, that of the Hadley cell, because the LIG leads to differential heating of the polar and tropical regions. For instance, ISM variability over the Tibetan Plateau is found to be more sensitive to the summer LIG between 44°N and the Himalayan barrier at 30°N than to the cross-equatorial LIG (Ramisch et al., 2016). A study from the West African monsoon region shows a strong link of intra-hemispheric summer LIG and the West African monsoon (Kuechler et al., 2018). Further, different seasonal changes of the LIG (summer LIG varies with the obliquity and winter LIG with precession) have a strong response to climatic changes during the past (Davis and Brewer, 2009). Winter and summer LIG are equally important drivers for the strengths, positions and areal extents of the STWJ and the ISM (Mohtadi et al., 2016; Ramisch et al., 2016). As the catchment area of the core location is influenced by both, ISM (summer) and STWJ (winter), the winter LIG of the mid-

high latitudes has to be considered. Fallah et al. (2017) found that the STWJ over Iran ranged between 35°N and about 54°N during the last 6 ka. During the winter months (here defined as December – February), the winter LIG (here defined as the gradient between 60°N-30°N) determines the southernmost position of the STWJ and the frequency of the WDs (Fallah et al., 2017). The summer LIG (here defined as the gradient between 30°N – equator as the Himalayan range acts an barrier, June-August) affects the strength and extension of the ISM over centennial to millennial timescales (Ramisch et al., 2016). The antagonism and relative magnitudes of the winter and summer LIGs influence the duration of the ISM, as well as STWJ and the frequency of the WDs. Therefore, changes in LIG have probably modulated seasonal length of the IM in the course of the Holocene.

The summer LIG reached its maximum between 8 and 7 ka BP, whereas the winter LIG peaked during the last 1 ka (Figure 2). The net effect of ISM weakening and STWJ strengthening, in combination with dry periods on the adjacent continent to the north, resulted in more frequent dust storms that increased aeolian input to the Arabian Sea during the Late Holocene (Burdanowitz et al., 2019). We suggest that the time period between 4.6 and 3 ka BP marks a transition from an ISM-dominated climate system towards one which is more influenced by the STWJ. A wavelet analysis of LIT MAR data (Figure 3) illustrates the shift of influences during the transition period with a distinct change of periodicities between ca 4.2 and 3.5 ka BP. The last 3.5 ka of the LIT MAR record is characterized by periodicities of about 512 to 1024 years, whereas periodicities of about 2048 to 3000 years were dominant between 7.5 to 3.5 ka BP. This change of periodicities around 3.5 ka BP indicates that the Makran region was more influenced by the mid-high latitude climate during the past 3.5 ka BP than during the Mid-Holocene. During the Mid-Holocene, in contrast, the ISM and the low latitude climate were dominant drivers in the Makran region and suppressed the influence of the Bond events. These findings corroborate the hypothesis of a teleconnection between the mid-high-latitude NH climate via the SWTJ and IWM in the NE Arabian Sea region (Böll et al., 2015; Munz et al., 2015, 2017). On a more regional scale, increasing influence of the STWJ coinciding with the transition phase is documented in several marine and terrestrial records from the Arabian Sea and Indian subcontinent (Giosan et al., 2018; Hou et al., 2016). Climate simulations for the last 6 ka over Iran have shown a southward shift of the STWJ during the Late Holocene due to increasing winter insolation (Fallah et al., 2017), suggesting a "tipping point" around 3 - 4 ka BP in the interaction of ISM and STWJ with waxing influence of the STWJ towards the present day. The "tipping point" corresponds to so-called neoglacial anomalies in the Makran region, when a strong winter monsoon and weak interhemispheric temperature contrast (30°N-30°S) were accompanied by a decrease in ISM precipitation (Giosan et al., 2018). Significant temperature oscillations over the last 3 ka were also recorded in the continental record of Lake Qinghai, but are commonly interpreted as an amplified response to volcanic and/or solar forcing (Hou et al., 2016). The manifestation of a meridional shift in STWJ is not restricted to the Arabian Sea region. A southward displacement of the STWJ during the Late Holocene has also been reported from the southern Alps in Europe, the Japan Sea and climate simulations for East Asia (Kong et al., 2017; Nagashima et al., 2013; Wirth et al., 2013), indicating teleconnections on a hemispheric spatial scale.

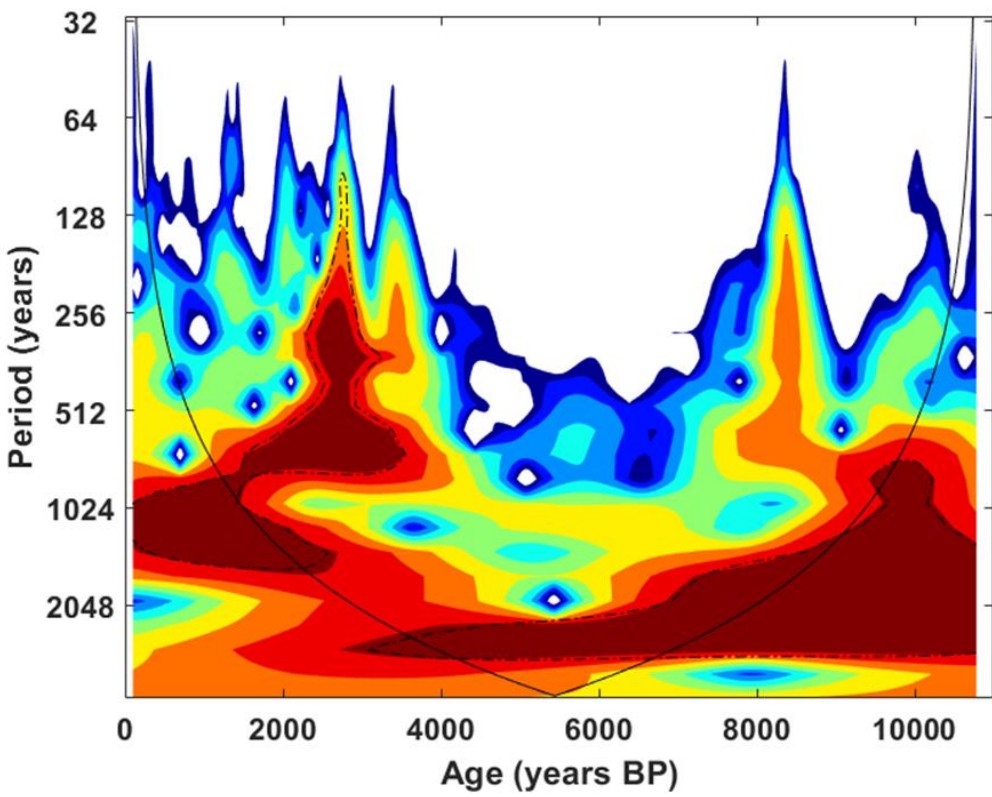

**Figure 3: Change of the periodicity in the lithogenic mass accumulation rates (LIT MAR) record. Wavelet power spectrum for LIT MAR of SO90-63KA using a Matlab Wavelet script (Torrence and Compo, 1998). The black line indicates the cone of influence, the dashed black line the 95% significance level. The periodicities clearly change in the time from 4.2 ka to 3.5 ka BP from ca. 2360 years during the Early and Mid-Holocene to ca. 590-710 and 1400 years during the Late Holocene.**

Several studies have suggested ENSO as an important driver modulating the climate in the IM realm during the Holocene (Banerji et al., 2020; Munz et al., 2017; Prasad et al., 2014, 2020; Srivastava et al., 2017). In general, an ENSO event is associated with reduced ISM precipitation (Gadgil, 2003). Even though the ENSO events have apparently increased over the last ca. 5000 years (Moy et al., 2002), roughly matching the timing of our observed transition period, we cannot find a correlation of our LIT MAR record with periods of strong ENSO events (not shown). Other studies have suggested that the relationship between IM precipitation and ENSO is not linear. Observations (1880 – 2005) imply that less than half of the severe droughts occurred during El Niño years, while the amount of precipitation was normal or above normal during other El Niño years (Rajeevan and Pai, 2007). A possible reason for this mismatch is the strong effect of the Indian Ocean Dipole (IOD). It can act as an amplifier or suppressor of the ENSO influence in the ISM realm (Ashok et al., 2001, 2004; Behera and Ratnam, 2018; Krishnaswamy et al., 2015; Ummenhofer et al., 2011). A positive IOD can counteract even a strong El Niño forcing leading to "normal" precipitation amounts (Ashok et al., 2004; Krishnaswamy et al., 2015; Ummenhofer et al., 2011). A recent study further suggested that a positive IOD shows a tripolar pattern over the IM realm, with above normal precipitation

in central India and below normal precipitation in northern and southern India (Behera and Ratnam, 2018). In contrast, a negative IOD creates a zonal pattern with above normal precipitation in western India and below normal precipitation in eastern India (Behera and Ratnam, 2018). In addition, these authors have linked a positive IOD to warm SSTs in the northern AS and vice versa. However, the impact of the IOD during the transition period is unknown due to a lack of paleorecords and reconstructions.

## 4.3 The environmental-anthropogenic interaction during the mid-Late Holocene

Widespread anthropogenic land use, such as deforestation or rice irrigation, and associated environmental change is recorded during the Early-Mid Holocene transition period (Petrie et al., 2017; Petrie and Bates, 2017). Especially rice irrigation, starting around 6-4 ka ago, has been suggested to have increased the atmospheric $CH_4$ concentration since about 5 ka BP (Figure 4) and thus had a small, but emblematic impact on climate (Ruddiman and Thomson, 2001). The rise of the Indus civilization began around 5.2 ka BP when general aridification decreased river discharges (Giosan et al., 2012). Since ca. 4 ka BP, increasing winter precipitation enabled the development of agriculture along the Indus valley due to less intense floods favoring urbanization (Giosan et al., 2012; Wright et al., 2008). The urban Harappan civilization collapsed towards the end of the climatic transition period when less frequent annual flooding occurred, which was the basis of agriculture. Most of the people migrated from the Indus valley to the Himalaya plains, whereas the remaining society turned into a post-urban society (Clift and Plumb, 2008; Giosan et al., 2012, 2018; Possehl, 1997).

The average chain length (ACL) of plant-wax derived long-chain *n*-alkanes varies in response to vegetation type and/or climatic conditions (Bush and McInerney, 2013; Carr et al., 2014; Meyers and Ishiwatari, 1993; Rao et al., 2011; Vogts et al., 2009). For instance, African savanna plants produce, on average, longer chain *n*-alkanes than rainforest plants (Rommerskirchen et al., 2006; Vogts et al., 2009). In the region of Indus civilization, natural vegetation dominated until agriculture proliferated (Giosan et al., 2012) driven by changing precipitation pattern due to the interplay of the position and strength of the STWJ and ISM (Ansari and Vink, 2007) (Figure 4). During the Mid-Holocene warm period, high SSTs are thought to have fueled the moisture transport to the adjacent continent, which is in line with our SST record (Figure 2). High average summer precipitation and/or more uniformly distributed precipitation during the different seasons favored the development of grasslands in southern Pakistan (Ansari and Vink, 2007; Ivory and Lézine, 2009). This may be reflected in a decrease in the $ACL_{27-33}$ between ca. 7 to 5 ka BP (Figure 4). The $ACL_{27-33}$ increased after the transition period, although the amplitude of $ACL_{27-33}$ variation is not very high, and decreasing river discharges and increasing aeolian input (Figure 2) indicate prevailing dry conditions after 3 ka BP. It is impossible to separate effects of anthropogenic modified vegetation or climatic driven vegetation changes on $ACL_{27-33}$ fluctuations as the $ACL_{27-33}$ reflects to whole vegetation community. However, the $ACL_{27-33}$ increase suggests that humans have started to influence the vegetation in the sediment source region and led to the decoupling of the climate signal and the $ACL_{27-33}$. This is in line with the coeval decoupling of the atmospheric $CH_4$ concentration and orbitally driven climate changes (Ruddiman and Thomson, 2001). These authors have suggested that inefficient early rice farming caused atmospheric $CH_4$ concentration to increase at around 5 ka BP. Rice agriculture began

around 4 ka BP on the Indian subcontinent and near the Indus River region, adding to the atmospheric $CH_4$ concentration (Fuller et al., 2011). Further, climatic changes during the Mid-Holocene were suggested to have changed the agricultural system in the region to a "multi-cropping" (summer/winter crops) system (Petrie and Bates, 2017; Wright et al., 2008). This clearly shows that human activities not only modified the vegetation, but also started to impact climate during this time. In a

380    modelling study land cover change associated with prehistoric cultures modified not only local climate, but climate on a broader hemispheric scale via teleconnections (Dallmeyer and Claussen, 2011). Since the transformation of natural into cultivated landscapes favors soil erosion, it is likely that this early human land-use change intensified the impact of the NAO and Bond events on the sedimentation in the NE AS during the late Holocene. The latter in turn documents a clear shift of the climatic system that was associated with the collapse of and deep crises of Late Bronze Age societies in the Mediterranean,

385    Middle East and East Asia.

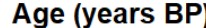

**Figure 4: Environmental and anthropogenic interactions illustrated by proxy records. a) different phases of the Indus civilization (Giosan et al., 2012) (L=Late, M=Mature, E=Early) and time periods of interpreted natural and anthropogenic modified vegetation (green bar and arrow) of core SO90-83KA in comparison to b) average chain length (ACL) of the *n*-alkanes homologues 27-33 (black line, 5 point running mean) and c) LIT MAR of SO90-63KA. The ISM humidity index (Herzschuh, 2006) (blue), CH4 concentration (red) reconstruction based on Greenland ice cores (Blunier et al., 1995) are also depicted in panel b and d) Bond events (numbered grey bars) based on the HSG drift-ice record from the North Atlantic (Bond et al., 2001).**

The climate transition period between 4.6 and 3 ka BP impacted not only the Indus civilization but also civilizations in Mesopotamia, the Eastern Mediterranean and China (Giosan et al., 2012, 2018; Kaniewski et al., 2013; Liu and Feng, 2012; Wang et al., 2016; Weiss et al., 1993). Today these regions are located slightly north of the northern border of the region influenced by the Asian monsoon (Wang et al., 2016), but where probably within the monsoon realm prior to the transition

period. The series of climatically triggered collapses started at about 4.4 ka BP in China and reached Mesopotamia and the Eastern Mediterranean between 3.1 and 3.2 ka BP (Giosan et al., 2012, 2018; Kaniewski et al., 2013; Liu and Feng, 2012; Wang et al., 2016). This indicates that this transition period was characterized by a southward shift of the ISM realm, which in combination with socio-economic responses turned into catastrophic changes for Late Bronze Age societies that thrived in the ISM/ STWJ transition zone.

## 5 Conclusions

Our study of reconstructed SST, LIT MAR and vegetation changes at the Makran coastal region suggests a climatic transition period between 4.6 – 3 ka BP from a low latitude to increased mid-high latitude influence via the STWJ. Prior to this transition period the region was strongly influenced by the ISM and the ITCZ. Our LIT MAR record shows an up to 4-fold increase after the transition period coincident with Bond events in the North Atlantic. We argue that after this period the North Atlantic signals of NAO and Bond events are transmitted to the Makran coast region via the southward shifted STWJ. This supports an earlier modelling study (Fallah et al., 2017) suggesting a southward shift of the STWJ and more winter precipitation near our study area in Iran between 4 and 3 ka BP, which the authors mark as a "tipping point" in that region. Besides a long term drying trend since the Mid-Holocene, this transition period and the associated change in the precipitation pattern (winter vs. summer precipitation) affected the settlements and agriculture of the Indus civilization (Giosan et al., 2012, 2018; Petrie and Bates, 2017). The temporal coincidence gives rise to the hypothesis that humans themselves became environmental drivers prior to the onset of the transition period (Fuller et al., 2011; Ruddiman and Thomson, 2001), instead of only being passive victims of climate change. Whether human impacts suffice to influence the global climate is an ongoing debate, but it is very likely that land cover changes associated with prehistoric cultures modified at least local climate (Dallmeyer and Claussen, 2011). Consequences included a weakened local hydrological cycle that favored desertification, especially in semi-arid regions.

## Data availability

The alkenone-based SSTs, the LIT MAR, the extended grain size used for the EM3 and *n*-alkane data sets of core SO90-63KA are available at PANGAEA data repository (https://doi.pangaea.de/10.1594/PANGAEA.925909). The previous grain size and elemental data of SO-90KA are available at PANGAEA data repository (https://doi.pangaea.de/10.1594/PANGAEA.900973).

## Author contribution

BG and KE designed the study. NB analyzed LIT MAR, alkenones and *n*-alkanes. TR analyzed grain sizes. NB, BG and TR interpreted the data. NB prepared the manuscript with input from all co-authors.

**Competing interests.**

The authors declare that they have no conflict of interest.

**Acknowledgements**

We thank F. Langenberg, M. Metzke, C. Staschok, L. Hilbig, L. Meiritz and Y. Akkul for analytical support. S. Beckmann is thanked for the technical support for *n*-alkane and alkenone analyses. We thank two anonymous reviewer for comments that helped to improve the manuscript. The study was supported by the German Federal Ministry of Education and Research (BMBF) as part of the CAHOL (Central Asian Holocene Climate, project number 03G0864A), a subproject of the research program CAME II (Central Asia: Climatic Tipping Points & Their Consequences, project number 03G0863G). This study is a contribution to the Cluster of Excellence 'CLICCS - Climate, Climatic Change, and Society', contribution to the Center for Earth System Research and Sustainability (CEN) of Universität Hamburg.

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
