# Peer review of "Signals of Holocene climate transition amplified by anthropogenic land-use changes in the Westerly-Indian Monsoon realm"

_Climate of the Past, 2020_

## Referee Comment (RC1) · Anonymous Referee #1 · 1 Feb 2021

Review of manuscript "Holocene climatic changes in the Westerly-Indian Monsoon realm and its anthropogenic impact" by Burdanowitz et al.. The main thrust of this manuscript is to use high-resolution proxy records from the NE-Arabian Sea reflecting various aspects of Holocene monsoonal changes in the region. The main new data series include Uk37 based SST estimates, alongside a number of proxy time series (Ti/Al, endmember modelled aeolian input and lithogenics mass accumulation rates) reflecting lithogenic input in the region with the latter being of central importance. Based on these data the authors conclude that the Arabian Sea region around 4.6-3kaBP became more sensitive to changes in the tropical westerly jet controlling climate in the region. Overall there may well be something interesting in this manuscript, but in the

current state it is not clear what this finding/story actually is. These are just a few problems. In my judgement, the biggest issue is the combination of results and discussion. In the absence of a dedicated results chapter the text rather selectively describes certain findings, whilst (largely) ignoring others. As an example, the authors place emphasis on relationship of the early Holocene rises in the LitMAR record (ignoring at this stage that there is a gap in the record near the time period being discussed) and Bond event 5. Later in the manuscript, again, it is being emphasised that there is a relation between the final Bond events and variability in the LitMAR record. What about the Bond cycles between 7.5. and 3.5kaBP. There is no relation in my view between these cycles and the LitMAR record, which in this interval has no obvious signal. What is this mismatch driven by, the alleged climatological connection between the Bond events and sedimentation in the Arabian Sea, or is the LitMAR proxy not sufficiently sensitive? With regard to the wavelet power spectrum, I am not convinced of the usefulness in this case, the main problem being the lack of a clearly visible signal in the LitMAR record between 7.5-3.5 kaBP. How does that affect the overall analysis? Also, it would help to inform the reader of the main findings based on this analysis (in the main text) rather than just alluding to change in frequency. There are quite a few statements that lack clarity regarding the implied change in the monsoon system and therefore appear contradictory. As an example in lines 154/5 there is this statement "This warm period encompasses the Mid-Holocene climate optimum period and is characterized by low LIT MAR and increasing fluvial input (Figure 2).." Would increasing fluvial input not entail higher lithogenic sedimentation rates? If so, how does this compare the overall low LitMAR record? Similar inconsistencies regarding the general state of the monsoon circulation can be found elsewhere in the manuscript. With regard to the LIG approach there is not sufficient justification provided why the chosen gradients are the most appropriate. There have been other approaches (on different time scales) such as by Reichart who has used a different gradient. There should be a better explanation as to the reasons for choosing the LIG's. Overall, there may well be something interesting in this paper. Currently, however, it lacks maturity and requires a substantial rewrite.

There, should be a better separation between results (all) and the interpretation. In addition, the discussion should be "closer" to the actual data. Large parts of the text read like and general discussion with a loose relation to the actual observations. More could be said.

---

## Referee Comment (RC2) · Anonymous Referee #2 · 15 Feb 2021

The paper submitted to Climate of the past by Burdanowitz et al. (Holocene climatic changes in the Westerly-Indian Monsoon realm and its anthropogenic impact), aims at providing new insights into how the potential interactions between ITCZ dynamics and Indian Monsoon, in the one hand, and Sub-Tropical Westerly Jets, in the other hand, may have driven orbital and millennial climate changes over the NE Indian monsoon area during the Holocene.

Although this issue is clearly an important one, the discussion does not successfully reach its objectives because the manuscript gives a feeling of confusion and ad-hoc argumentation in many places. The discussion is based upon six main sets of data ob-

tained in core SO90-63KA, among which only three are (apparently) published here for the first time, and are given a thorough description in the method chapter (Lithological Mass Accumulation Rates, Uk37'-SST, and the average chain length of the n-alkanes homologues 27-33). The other data having been published elsewhere, the readers are left with only minimal to no piece of information about how these proxies were obtained and/or are interpreted. The lack of information is detrimental to a clear understanding of the authors' arguments. For instance, Ti/Al is interpreted, here, as being positively associated to higher river contributions (fig. 2), which is opposite to what has been concluded for sediments from the tropical Atlantic (Govin et al., 2012). Why is that so? Clearly elemental ratios should be interpreted in the light of regional/local rock sources, transport mechanisms, etc.. The basis for the interpretation of Ti/Al in the Arabian Sea should be summarized somewhere in the method chapter. I've had the same kind of issues with basically all the proxies used in the manuscript. How EM3 record was obtained? What does it mean? What about the planktic DNA ? Etc. I was also surprised that LIT-MAR is given such an importance, given the fact that the core was retrieved thirty years ago. It is very likely that wet weights obtained for calculating DBD have been largely modified by evaporation since the core retrieval. The authors themselves point out that some part of the core completely dried out. To which extent can this drying impact the DBD and does that have a significant effect on LIT-MAR estimates?

Because the data are not presented in a dedicated "result" chapter prior to the discussion, one has the impression that the authors build their interpretation by jumping from one proxy to another, highlighting the patterns that suit their hypothesis. Every now and then they even appeal to important data, not presented in the article. This is the case for G. ruber ïẠď18O, which they cite to strengthen their argument on past changes in precipitation and river runoff. If the G. ruber ïẠď18O record has already been published and can bring interesting pieces of information about salinity (precipitation, runoff) and temperature changes, it should be shown in the present manuscript and thoroughly compared with Uk37' SST and Ti/Al records. . . Not used to highlight just a specific feature observed in the Ti/Al record. The lack of a thorough discussion

about the data also results in some key features of the records not given the proper attention. What about, for instance, the long-term change in the Ti/Al record, which amplitude contrasts with the small amplitude of the millennial-scale variations? Why is the LIT-MAR record showing a rather opposite mode of variability (ie. lack of long-term mode of variation, short episodes of higher MAR)? Why are the Ti/Al and LIT-MAR so evidently decoupled from one another?

All the above questions about proxy interpretation and comparison should be addressed in the manuscript. Thus, the manuscript needs a thorough rewriting with (i) added pieces of information about the proxies signification and interpretation; (ii) and a dedicated "results" chapter in which records are presented thoroughly before being referred to in the discussion. This should serve as a basis for a more organized and clearly argued discussion.
* * *

---

## Author Comment (AC1) · 17 Mar 2021

First of all, we thank the two anonymous referees for their constructive comments which helped us to improve our manuscript. In the following we give a step-by-step response to the referee comments (given in black, our responses are given in blue).

RC1-1: Review of manuscript "Holocene climatic changes in the Westerly-Indian Monsoon realm and its anthropogenic impact" by Burdanowitz et al.. The main thrust of this manuscript is to use high-resolution proxy records from the NE-Arabian Sea reflecting various aspects of Holocene monsoonal changes in the region. The main new data series include Uk37 based SST estimates, alongside a number of proxy time series (Ti/Al, endmember modelled aeolian input and lithogenics mass accumulation rates) reflecting lithogenic input in the region with the latter being of central importance. Based on these data the authors conclude that the Arabian Sea region around 4.6-3kaBP became more sensitive to changes in the tropical westerly jet controlling climate in the region. Overall there may well be something interesting in this manuscript, but in the current state it is not clear what this finding/story actually is. These are just a few problems.

Response: We thank the anonymous referee #1 for his/her constructive comments on our manuscript.

RC1-2: In my judgement, the biggest issue is the combination of results and discussion. In the absence of a dedicated results chapter the text rather selectively describes certain findings, whilst (largely) ignoring others.

Response: We agree with referee #1 and also referee #2 who have raised concerns about the combination of results and discussion sections. We have rewritten the "results & discussion" part and have divided it into two separate sections. We also agree with both referees that this highly increases the readability and flow of arguments of our manuscript.

RC1-3: As an example, the authors place emphasis on relationship of the early Holocene rises in the LitMAR record (ignoring at this stage that there is a gap in the record near the time period being discussed) and Bond event 5. Later in the manuscript, again, it is being emphasised that there is a relation between the final Bond events and variability in the LitMAR record. What about the Bond cycles between 7.5. and 3.5kaBP. There is no relation in my view between these cycles and the LitMAR record, which in this interval has no obvious signal. What is this mismatch driven by, the alleged climatological connection between the Bond events and sedimentation in the Arabian Sea, or is the LitMAR proxy not sufficiently sensitive?

Response: We thank the referee #1 for stressing this mismatch, which was not properly explained in our manuscript. We are aware that we cannot exclude enhanced LIT MAR within the gap period (9.4 to 8.5 ka BP). However, between the gap period and the enhanced LIT MAR around 8.3 to 8.2 ka BP two low data points suggest to us that this period of enhanced LIT MAR may a short event caused by high soil erosion around the 8.2 ka event. We have added this information to the manuscript. The referee #1 is correct that there is no apparent relationship between the Bond cycles in the time interval 7.5 to 3.5 ka BP and the LIT MAR. This is in our interpretation an additional support for our hypothesis that the period between 4.6 and 3.0 ka is a transition period from low to mid-high latitude influence. We argue that the Bond event signal and the climate variability in

the North Atlantic, respectively, have a stronger teleconnection to the Makran region, when STWJ is located more to the south. Therefore, the Makran region is more influenced by the mid-high latitude climate during the past 3.5 ka than during the Mid-Holocene (ca. 7.5 to 3.5 ka BP). Thus, the ISM and the low latitude climate were dominant drivers in the Makran region and suppressed the influence of the Bond events during the Mid-Holocene. We stressed the teleconnection between the mid-high latitude climate and the Makran region in the previous version of the manuscript (lines 210 -213):

"*We suggest that the time period between 4.6 and 3 ka BP marks a transition from an ISM-dominated climate system towards one which is more influenced by the STWJ. This strengthened the teleconnection between the Makran coast and climate variability in the North Atlantic, which is most visible by the link between the LIT MAR record and the Bond events since the end of the time period and associated fall the Indus civilization (Figure 2).*"

and (lines 284 - 287)

"*Since the transformation of natural into cultivated landscapes favors soil erosion, it is likely that this early human land-use change intensified the impact of the NAO and Bond events on the sedimentation in the NE AS during the late Holocene. The latter in turn documents a clear shift of the climatic system that was associated with the collapse of and deep crises of Late Bronze Age societies in the Mediterranean, Middle East and East Asia.*"

However, since a clear discussion about the teleconnection of Bond events and LIT MAR during the Mid-Holocene period was lacking in the previous version of the manuscript, we have now included it in the discussion.

RC1-4: With regard to the wavelet power spectrum, I am not convinced of the usefulness in this case, the main problem being the lack of a clearly visible signal in the LitMAR record between 7.5-3.5 kaBP. How does that affect the overall analysis? Also, it would help to inform the reader of the main findings based on this analysis (in the main text) rather than just alluding to change in frequency.

Response: We agree with referee #1 that we have to include the main findings of the wavelet power spectrum analysis to better understand the visible and hidden signals. For this analysis we used a Matlab script and interpolated our data set to achieve a temporal resolution of about 15.6 years. This is the minimum time step between two samples and also covers the gap between 9.4 and 8.5 ka BP. The advantage of the wavelet power spectrum is that it shows how the periodicities differ/change during the time. For instance, there are time periods that do not show a specific periodicity signal. In our LIT MAR record, this is the case for the periodicity of about 512 to 1024 years observed during the last 3.5 ka compared to the prevalent periodicity of about 2048 to 3000 years in the time period between 7.5 – 3.5 ka BP. The absence of these shorter periodicities during the Mid-Holocene is a further evidence that Bond events (cycle length roughly of about 1500 years) and, therefore, mid-high latitude climate have played a minor role in the Makran region during that time. We have added further description to the results and the discussion section to highlight the findings of the wavelet power spectrum analysis.

RC1-5: There are quite a few statements that lack clarity regarding the implied change in the monsoon system and therefore appear contradictory. As an example in lines 154/5

there is this statement "This warm period encompasses the Mid-Holocene climate optimum period and is characterized by low LIT MAR and increasing fluvial input (Figure 2).." Would increasing fluvial input not entail higher lithogenic sedimentation rates? If so, how does this compare the overall low LitMAR record?

Response: We agree with referee #1 that some statements were not clearly explained. As referee #2 has the same issue regarding the contrasting behaviour of fluvial input (Ti/Al ratios) and LIT MAR (indicator for soil erosion), we have added further text to the discussion to explain our reasoning. The LIT MAR is decoupled from the Ti/Al records as it is an indicator for soil erosion and is not necessarily coupled to only fluvial input, but also to aeolian input. Arid conditions are necessary for enhanced soil erosion to occur, because vegetation prevents soil erosion. But also during arid phases there can be flood events triggered by precipitation events, that cause a strong fluvial sediment transport. This is, for instance, also evident in the Namaqualand mudbelt sediments offshore western South Africa (Herrmann et al. 2016). We posit, that the combination of pronounced soil erosion, flood events and strong winds during arid phases led to the high LIT MAR in the late Holocene part of our record.

RC1-6: Similar inconsistencies regarding the general state of the monsoon circulation can be found elsewhere in the manuscript. With regard to the LIG approach there is not sufficient justification provided why the chosen gradients are the most appropriate. There have been other approaches (on different time scales) such as by Reichart who has used a different gradient. There should be a better explanation as to the reasons for choosing the LIG's.

Response: We agree with referee #1 that the different approaches and concepts regarding the latitudinal insolation gradient (LIG) are not properly explained in the manuscript and have now added further information. In general, there are two different concepts for the LIG. Reichart (1997) and Bosmans et al. (2015) argued that the boreal summer cross-equatorial insolation gradient plays an important role for the inter-hemispheric moisture transport as well as the extent of the Hadley cell and may drive glacial-interglacial variability. The other concept by Davis & Brewer (2009) highlights the importance of intra-hemispheric insolation gradients during summer (driven by obliquity) and winter (driven by precession). These authors stated that intra-hemispheric LIG influences the strength of the monsoon system and its most poleward position as well as the position of the Hadley cell. For our study, we were interested in the northernmost position of the ISM and the southernmost position of the STWJ, because the core location is influenced by both systems. A study from the West African Monsoon region found a strong link of intra-hemispheric summer LIG (60° – 30°N) and strength of the West African monsoon (Küchler et al. 2018). Ramisch et al. (2016) found that the ISM variability over the Tibetan Plateau may be sensitive to the summer LIG between 44°N and the Himalayan barrier at 30°N rather than cross-equatorial LIG. Therefore, we have chosen the summer LIG between the equator and 30°N south of the Himalayan barrier (Ramisch et al., 2016). The STWJ reaches its southernmost position during the winter in the study area and the position of the STWJ ranged from 35°N and ~54°N during the last 6 ka (Fallah et al., 2017). Therefore, we have chosen the winter LIG between 30°N and 60°N.

RC1-7: Overall, there may well be something interesting in this paper. Currently, however, it lacks maturity and requires a substantial rewrite. There, should be a better separation between results (all) and the interpretation. In addition, the discussion should be "closer"

to the actual data. Large parts of the text read like and general discussion with a loose relation to the actual observations. More could be said.

Response: We thank the referee #1 for his/her constructive comments which help us to improve the manuscript. Based on these comments we have separated the "results & discussion" part into "results" and "discussion" chapters. In addition, we have added information about the various proxies employed and their links to mid-high latitude climate and Bond events, respectively. We have rewritten parts of the discussion to better support our findings. We have also decided add additional data relevant for this manuscript as supplementary material. This includes results of grain size analyses and end-member modelling analyses as well as data on the $n$-alkanes ($CPI_{27-33}$, $ACL_{27-33}$ and $n$-alkane distribution).

---

## Author Comment (AC2) · 17 Mar 2021

First of all, we thank the anonymous referee for constructive comments which helped us to improves our manuscript. In the following, we give a step-by-step response to the referee comments. The referee comments are given in black, our responses are given in blue.

RC2-1: The paper submitted to Climate of the past by Burdanowitz et al. (Holocene climatic changes in the Westerly-Indian Monsoon realm and its anthropogenic impact), aims at providing new insights into how the potential interactions between ITCZ dynamics and Indian Monsoon, in the one hand, and Sub-Tropical Westerly Jets, in the other hand, may have driven orbital and millennial climate changes over the NE Indian monsoon area during the Holocene. Although this issue is clearly an important one, the discussion does not successfully reach its objectives because the manuscript gives a feeling of confusion and ad-hoc argumentation in many places.

Response: We thank the anonymous referee #2 for his/her constructive comments on our manuscript.

RC2-2: The discussion is based upon six main sets of data obtained in core SO90-63KA, among which only three are (apparently) published here for the first time, and are given a thorough description in the method chapter (Lithological Mass Accumulation Rates, Uk37'-SST, and the average chain length of the n-alkanes homologues 27-33). The other data having been published elsewhere, the readers are left with only minimal to no piece of information about how these proxies were obtained and/or are interpreted. The lack of information is detrimental to a clear understanding of the authors' arguments.

Response: We agree with the referee #2 that we need to include more information about the data published earlier. We have added further information about the extended EM3 record and the previously published Ti/Al record in the method section and discussion, respectively. As also mentioned in the response to RC2-4 below, we realized that the citation of Giosan et al. (2018) for the Indus 11C record is missing in the figure caption and have added it now in the revised version of the manuscript. We are sorry that this led to confusion.

RC2-3: For instance, Ti/Al is interpreted, here, as being positively associated to higher river contributions (fig. 2), which is opposite to what has been concluded for sediments from the tropical Atlantic (Govin et al., 2012). Why is that so? Clearly elemental ratios should be interpreted in the light of regional/local rock sources, transport mechanisms, etc.. The basis for the interpretation of Ti/Al in the Arabian Sea should be summarized somewhere in the method chapter.

Response: Our interpretation of the Ti/Al record is based on earlier analyses (Lückge et al. 2001) of elemental ratios of marine sediments nearby our core site as well as fluvial sediments along the Hingol River, which adds to the sedimentation at our core site. They found high Ti/Al ratios in the fluvially deposited sediments along the Hingol River and also found high Ti/Al ratios in thicker varves of marine sediments in the NE Arabian Sea than in thinner varves. Lückge et al. (2001) concluded that high Ti/Al ratios indicate increased transport energy and therefore more intense fluvial discharge than low Ti/Al ratios. As mentioned in the response to RC2-2, we have added further information about the Ti/Al record and its interpretation in the method section.

RC2-4: I've had the same kind of issues with basically all the proxies used in the manuscript. How EM3 record was obtained? What does it mean? What about the planktic DNA ? Etc.

Response: As for the Ti/Al record we have included further information about the EM3 record, which is based on grain size endmember modelling, in the method section (see also response to RC2-2). Concerning the Indus 11C SST record based on planktic DNA (Giosan et al., 2018) we realized that the citation "Giosan et al., (2018)" was missing in the caption of Figure 2 b) and have added it in the revised version of the manuscript.

RC2-5: I was also surprised that LIT-MAR is given such an importance, given the fact that the core was retrieved thirty years ago. It is very likely that wet weights obtained for calculating DBD have been largely modified by evaporation since the core retrieval. The authors themselves point out that some part of the core completely dried out. To which extent can this drying impact the DBD and does that have a significant effect on LIT-MAR estimates?

Response: The referee #2 has a valid point concerning about the impact of drying of core material on dry bulk density and LIT MAR. However, the core was well preserved except the core section between 547 – 597 cm. This section was dried-out and had shrunken by about 10%. During the core sampling we compared each core section with existing radiographies of this core and found that there was no apparent depth bias. Of course, there may be some evaporation effects for the whole core after thirty years and it is likely that there is a bias of the DBD. But as all core parts, except for 547 – 597 cm, were in the unchanged condition it is unlikely that differences in the LIT MAR are due to punctuated drying artifacts.

RC2-6: Because the data are not presented in a dedicated "result" chapter prior to the discussion, one has the impression that the authors build their interpretation by jumping from one proxy to another, highlighting the patterns that suit their hypothesis. Every now and then they even appeal to important data, not presented in the article.

Response: We agree with both referees who objected to combined results and discussion sections. We have rewritten the "results & discussion" part and have divided it into two separate sections. We also agree with both referees that this highly increases the readability of our manuscript and makes it much easier to follow our arguments.

RC2-7: This is the case for G. ruber δ18O, which they cite to strengthen their argument on past changes in precipitation and river runoff. If the G. ruber δ18O record has already been published and can bring interesting pieces of information about salinity (precipitation, runoff) and temperature changes, it should be shown in the present manuscript and thoroughly compared with Uk37' SST and Ti/Al records . . . Not used to highlight just a specific feature observed in the Ti/Al record.

Response: We agree with referee #2 and have added the G. ruber δ18O record of Staubwasser et al. (2002, 2003) to figure 2. We decided against including the G. ruber δ18O record of Giesche et al. (2019) in figure 2, because Giesche et al. (2019) reported δ18O of G. ruber from a core interval corresponding to the time period of 3.0 to 5.4 ka BP of a different size fraction (400-500 µm) than Staubwasser et al. (2003, 315-400 µm). Although there is an offset by 0.23 ‰ between these two data sets, the 210-year smoothed

trends of both records are in good agreement. Due to the short time period analysed by Giesche et al. (2019) this record was not included in the figure, bus has now been cited.

New figure 2:

[Figure]

RC2-8: The lack of a thorough discussion about the data also results in some key features of the records not given the proper attention. What about, for instance, the long-term change in the Ti/Al record, which amplitude contrasts with the small amplitude of the millennial-scale variations? Why is the LIT-MAR record showing a rather opposite mode of variability (ie. lack of long-term mode of variation, short episodes of higher MAR)? Why are the Ti/Al and LIT-MAR so evidently decoupled from one another?

Response: We agree with the reviewer that the discussion of some results lacked proper depth. The long-term trend of the Ti/Al record was already described by Burdanowitz et al. (2019), but we have added a more detailed description to this manuscript. The strong increase of Ti/Al ratios between about 11 ka to 9 ka BP may be due to a combination of the strengthening of the ISM and rising sea-level causing a more proximal core position and an altered fluvial input. During the Early and Mid-Holocene the Ti/Al record was mainly affected by Indian Summer Monsoon precipitation. From the Mid- to Late Holocene the influence of westerly induced precipitation on the Ti/Al gradually increased compared to the influence of the Indian Summer Monsoon. This combined Indian Summer Monsoon and westerly influence resulted in increased precipitation and therefore increased fluvial input, peaking during the Mid-Holocene. Conversely, a general aridification trend led to generally decreased fluvial input during the late Holocene. A spectral analysis of the Ti/Al record has shown a cycle of about 1425 years, similar to the Bond events, that is superimposed on this long-term trend (Burdanowitz et al., 2019). The LIT MAR is decoupled from the Ti/Al records as it is an indicator for soil erosion and is not necessarily coupled to only fluvial input, but also to aeolian input. Furthermore, more arid conditions promote soil erosion, because vegetation that prevents soil erosion is on the retreat. However, flood events resulting in a strong fluvial sediment transport also occur during arid phases. This is, for instance, also found in the Namaqualand mudbelt sediments offshore western South Africa (Herrmann et al. 2016). We assume that the combination of stronger soil erosion, flood events and also strong winds led to the higher LIT MAR during the arid phases of the Late Holocene in our record (see reply to RC1-5).

RC2-9: All the above questions about proxy interpretation and comparison should be addressed in the manuscript.
Thus, the manuscript needs a thorough rewriting with (i)added pieces of information about the proxies signification and interpretation; (ii) and a dedicated "results" chapter in which records are presented thoroughly before being referred to in the discussion. This should serve as a basis for a more organized and clearly argued discussion.

Response: We thank the referee #2 for his/her comments that helped to significantly improve our manuscript. As mentioned in the previous responses, we have added previously missing information on rationale for proxies to the manuscript and also as supplementary material, and have now divided the "results & discussion" chapter into two chapters.

---

## Referee Report (RR1)

The manuscript has been clearly improved, both in terms of its structure (with a "data results" section that was missing in the original version) and content (with a clarified and much improved discussion).

Yet, I still have some questions/commentaries and suggest that the manuscript could be accepted but with additional minor improvements.

**1/ LIT MAR:** Considering the key importance of the LIT MAR record in the discussion, this proxy deserves an in-depth presentation in the manuscript. Instead of just providing the final, computed LIT MAR record, I would like to see also a figure with the bulk data that have been used to construct this record, namely: (i) the dry bulk density record, (ii) the non-carbonate % and (iii) the sedimentation rate record derived from the numerous [14]C dates (with the location of those dates along the core). This could be done in the main manuscript, or in the supplementary material.

Not only should the bulk data for LIT MAR be provided to the readers, but they should also deserve an in-depth discussion too.
It is striking that some of the main LIT MAR events described in the text (eg. LIT MAR peaks at ca 8.4 ka, ca 3.4 ka and ca 2.8 ka) are just defined by 1 or 2 data points only.. I would be interested to know what explains those extremely narrow and high amplitude changes (do they reflect peaks in sedimentation rate and a potential issue with dating? Do they reflect peaks in DBD or peaks in the lithogenic material %? Do they result from a combination of those three elements?). How confident can we be regarding those peculiar peaks? How sensitive is the Wavelet Analysis to the presence of such short, high amplitude peaks?

**3/ Line 143**: the authors state that the last 3 ka show more variable SST that before. Couldn't this be - at least partially - attributed to the fact that the temporal resolution is also higher over the top 3 ka than in some older parts of the record ? I counted, for instance, about 20 data points in the SST record over the 4.5-6.5 ka interval, but ~ 40 points over the 0-2 ka time interval, thus twice as much. Change in sampling resolution should have an effect on our capacity to resolve rapid SST variations. The d18O record, which displays a more constant sampling resolution along the core than the SST record, suggests a relatively similar/invariant type of periodicity across the entire Holocene. Was this record run through Wavelet Analysis? How this compares with the LIT MAR spectral evolution?

**4/ Wavelet analysis**: I wonder why wavelet analyses were not performed/shown for the other proxies as well. Is LIT MAR the only proxy that reveals a shift in periodicity across the Holocene? Shouldn't we expect to see also a change in periodicity in the EM3 record, for instance? Is it seen when performing wavelet analysis?

**5/ Figure 2**: Is the "factor 1 – SST" (figure 2b; Giosan et al.) used and discussed in the text ?? I couldn't find a call to this figure 2b in the text.

---

## Author Response (AR2)

First of all, we thank the two anonymous referees for their constructive comments which helped us to improve our manuscript. In the following we give a step-by-step response to the referee comments (given in black, our responses are given in blue).

RC1-1: Review of revised manuscript "Signals of Holocene climate transition amplified by anthropogenic land-use changes in the Westerly-Indian Monsoon realm" by Burdanowitz et al.. The main thrust of this manuscript is to use high-resolution proxy records from the NE-Arabian Sea reflecting various aspects of Holocene monsoonal changes in the region. The main new data series include Uk37 based SST estimates, alongside a number of proxy time series (Ti/Al, endmember modelled aeolian input and lithogenics mass accumulation rates) reflecting lithogenic input in the region with the latter being of central importance. Based on these data the authors conclude that the Arabian Sea region around 4.6-3kaBP became more sensitive to changes in the tropical westerly jet controlling climate in the region. In addition, early agricultural activities in the region, may be reflected in the data. Compared to the initial version of the manuscript, there are areas of improvement in the revised manuscript. There are, however, still a number of issues which prevent recommending publication at this stage. Below there is a list of comments related to specific sections of the manuscript (line referencing based on version with track changes being highlighted). More generally, there are still too many occasions where the aspects focussed on the discussion are not properly described in the results sections and there seems to be some randomness with regard which events to discuss, despite the undiscussed events being of the same size. The results section should be expanded (including all the purely descriptive elements from the discussion). Also, the entire ENSO angle is rather unconvincing. It may help, in a revised version to remove this link and focus more on the change in vegetation.

The list below is in random order or importance/gravity:

Response: We thank the anonymous referee #1 for his/her constructive comments on our manuscript.

RC1-2: Lines 40-42 Can any of these things be proven. WD disturbances on which time scale do they occur? Can this be resolved in the existing data. How exactly are, probably short term changes in the STWJ affecting the ISM? This should be better explained.

Response: We have extended this part including a recent tracking algorithm study using ERA interim data (Hunt et al. 2018). The authors found 6-7 WDs per month during the past 36 years, with precipitation rates of about 3 mm/day and 40 mm/day for 95 % and 5 %, respectively. They also found a higher frequency of the WDs with a more southern position of the STWJ. We therefore think, that a latitudinal change of the STWJ during the past should have had a strong impact on the frequency of WDs and the amount of winter precipitation over Pakistan and India. We have added following to introduction of the manuscript: "*A tracking algorithm of ERA-interim data found about 3000 WDs (6 – 7 per month) between 1979 and 2016 in the region of Pakistan and India. About five % of the WD had precipitation rates of about 40 mm/day and the rest had about 3 mm/day (Hunt et al., 2018). Further, the frequency of WDs in Pakistan and India during the winter months was higher when the position of the STWJ was shifted to the south (Hunt et al., 2018). Therefore, a latitudinal change in the position of the STWJ during the past should have had a strong impact on the frequency of WDs and the amount of winter precipitation over that region.*"

RC1-3: Lines 130-131: It is not clear how this is supposed to work. First, what is the average and best resolution of the record. Second, there needs to be a much better explanation why it

is justified (if it is) to "invent" data in a part of the record that does not have any measured data?

Response: It is necessary that the time steps are uniformly distributed to perform the wavelet power spectrum analysis using the Matlab Wavelet script by Torrence & Compo (1998). For that reason, we have used the highest temporal resolution of about 15.6 years existing between to samples and interpolated the data of the LIT MAR record to this time step. However, we are aware that this will lead to a high uncertainty, especially in the core region between 9.4. and 8.5 ka BP, where no data for LIT MAR exists. We have added further explanation to the manuscript: "*As uniformly distributed time steps are necessary to perform* the analysis we interpolated our data set to achieve a temporal resolution of about 15.6 years. This is the minimum time step between two samples and also covers the gap between 9.4 and 8.5 ka BP. *We are aware that the uncertainty may be high, especially during the gap period where no data for LIT MAR exist.*"

RC1-4: Line 143: 93 grain size classes between 10 and 3 mm??? - are there no small size classes in the sediments as would need to be expected in this region of the world ocean?

Response: There is a missing unit. It should be "between 10 **nm** and 3 mm". We have corrected it in the manuscript.

RC1-5: Line151-152: "High SSTs were registered between 8.1 and 5.2 ka BP during the Mid-Holocene." This is not correct. There is a short term cool period centred at ~7.8 Ka BP. Also, how is the averaging done. Is this just, as stated, a five point moving average (without any time control) or a time controlled filtering (it should be the latter).

Response: The referee #1 is correct that there is a short-term cool period about 7.8 ka BP visible with about 0.5°C cooler temperatures than before and after. However, the uncertainty of the Uk37 based SST record are about 1°C, which is why we have mainly concentrated on the time periods with a stronger and more rapidly emerging temperature contrast. We have used a simple five point moving average for both, SST and age.

RC1-6: Lines 217-220 The link between the 8.2ka event and the G. ruber isotope data is unclear. Also, the Ti/Al ratio change is small and not quite in phase (it seems). This sections would benefit from a rewording.

Response: We have rephrased this section and included the dates of changes of each proxy described. However, we are aware that there is no strong evidence of the 8.2 ka event in the core.

RC1-7: line 245 "indicated by..." The end of this sentence does not make sense.

Response: The referee #1 is correct. There is a left over. We have rephrased this part to "*which is indicated by salinity sensitive foraminifera*".

RC1-8: lines 248-250 The entire discussion surrounding SST/d18O-data is based on marginally significant data (i.e. most of the signal in either the Uk37 and the d18O ruber data

is within the uncertainties for either method). There is some change in those records, but the discussion should also reflect that this is based on a small signal.

Response: We agree with referee #1 and have added the following sentences to the discussion *"Although considering both variations of the SST and δ18O records may be partly within the methods uncertainties, we are convinced that they show a climatic signal. They are independent of each other and, in addition, the EM3 record shows a rapid change during that time."*

RC1-9: Line 253 It is unclear how a climate change in one direction leads to a two (opposite) changes in the early civilizations. Better to remove the link from the study or explain better (interpret jointly with other data in a section on anthropogenic change?).

Response: We follow the suggestion of referee #1 of removing the link to the study and deleted *"and coincides with the rise and fall of the Indus civilization (Wright et al., 2008)."*

RC1-10: Line 257 The statement is not correct. The LitMar data over the last 3ka are generally low with some superimposed spikes.

Response: We agree with referee #1 that the LIT MAR is not "high and variable" like the SSTs. To avoid confusion and misunderstanding we have rephrased the section to *""… high and variable SSTs, highly variable LIT MAR with strong peaks, and...".*

RC1-11: Lines 278 and following: The entire argument surrounding cross-equatorial insolation gradient as a driver for short term change in the record is weak. As the authors noted with regard to summer insolation, smooth changes in the latter are very unlikely to be the main cause for short term change. The gradient records are equally smooth and it is therefore not clear how these help to explain short term change in core SO90-63KA.

Response: We totally agree with referee #1 that the smooth changes of the gradients alone are not the cause for the short-term changes in the record. However, as we noted in the manuscript there is a tipping period between 4.6 and 3.5 ka BP marking a transition period from an ISM dominated climate system towards one which is more influenced by STWJ. This is a cause of the different seasonal changing insolation gradients. Therefore, a stronger teleconnection to the North Atlantic climate and Bond events (short term) are found for the last 3.5 ka BP in our record. To add clarity, we have added *"**This may be the cause of the different seasonal (summer and winter) LIG changes and** strengthened the teleconnection between the Makran coastal climate and climate variability in the North Atlantic, which is best illustrated by the link between the LIT MAR record and the Bond events during the last 3.5 ka (Figure 2)."* to the manuscript.

RC1-12: Lines 313-317 It is unclear which period is being referred to. In the first sentence a change between 4.2 and 3.5 ka is introduced. Thereafter, suddenly there is a reference to the period 7.5-3.5 kaBP -which is it?

Response: We have specified and rephrased this part to *"This **change of periodicities around 3.5 ka BP** indicates that the Makran region was more influenced by the mid-high latitude climate during the past 3.5 ka BP than during the Mid-Holocene.".*

RC1-13: Lines 338-354 This, again, is an unconvincing section. My understanding is that the authors try to discuss a non-finding, i.e. that there is not signal in the data that matches the ENSO signal. It would be better to remove the ENSO link entirely. First, in modern climate the link between ENSO and the ISM strength is complicated and not linear. Second, the change happens on times scales beyond the scope of the study and thirdly, the data of core SO90-63KA do not support a link. It therefore does not seem to make much sense to include a rather long discussion surrounding ENSO in the manuscript.

Response: The referee #1 is absolutely right that there is no link to ENSO as we have described it in our manuscript. We are also aware, that there is an ongoing debate for the present and also the past. Although there is no link between ENSO and our data, there are paleo-studies within the IM realm showing a link to ENSO (see lines 319 -320 "*Several studies have suggested ENSO as an important driver modulating the climate in the IM realm during the Holocene (Banerji et al., 2020; Munz et al., 2017; Prasad et al., 2014, 2020; Srivastava et al., 2017).*" We think, that it is not only important to describe possible links to climate drivers, such as ENSO or NAO, supported by data but also when there is no support by data.

RC1-14: As indicated above, the manuscript has improved but it is still not in a state supporting publication at this stage. A further revision is recommended.

Response: We thank the referee for his/her critical comments and have carefully and critically revised our manuscript based on the referee's comments.

RC2-1: The manuscript has been clearly improved, both in terms of its structure (with a "data results" section that was missing in the original version) and content (with a clarified and much improved discussion).

Yet, I still have some questions/commentaries and suggest that the manuscript could be accepted but with additional minor improvements.

Response: We thank the anonymous referee #2 for his/her constructive comments on our manuscript.

RC2-2: LIT MAR: Considering the key importance of the LIT MAR record in the discussion, this proxy deserves an in-depth presentation in the manuscript. Instead of just providing the final, computed LIT MAR record, I would like to see also a figure with the bulk data that have been used to construct this record, namely: (i) the dry bulk density record, (ii) the non-carbonate % and (iii) the sedimentation rate record derived from the numerous 14C dates (with the location of those dates along the core). This could be done in the main manuscript, or in the supplementary material.

Response: We agree with referee #2 and have added the lithogenic content, sedimentation rate and dry bulk density to the supplementary material (Figure S1, see answer to RC2-3). We also have included the LIT MAR record and all [14]C dates in Figure S1.

RC2-3: Not only should the bulk data for LIT MAR be provided to the readers, but they should also deserve an in-depth discussion too. It is striking that some of the main LIT MAR events described in the text (eg. LIT MAR peaks at ca 8.4 ka, ca 3.4 ka and ca 2.8 ka) are just defined

by 1 or 2 data points only.. I would be interested to know what explains those extremely narrow and high amplitude changes (do they reflect peaks in sedimentation rate and a potential issue with dating? Do they reflect peaks in DBD or peaks in the lithogenic material %? Do they result from a combination of those three elements?). How confident can we be regarding those peculiar peaks? How sensitive is the Wavelet Analysis to the presence of such short, high amplitude peaks?

Response: We have followed the referee's suggestion and have provided further discussion regarding the LIT MAR record. The LIT MAR record is mainly controlled by the sedimentation rates and less by the dry bulk density and the lithogenic content itself (see Figure S1). This is the case for LIT MAR peaks around 8.4 ka, 3.4 ka and 2.8 ka BP. The LIT MAR peak around 1.4 ka BP coincides with slightly increased sedimentation rates and increased bulk density.

We have added following sentence to the results section (3.3): "*LIT MAR is mostly controlled by the sedimentation rate of SO90-63KA and less by the dry bulk density and the content of lithogenic material (Fig. S1), although the lithogenic material content slightly increases from the Mid- to Late Holocene.*" and to the discussion: " *However, this signal is not very pronounced in the SO90-63KA record and the peak around 8.4 ka BP in the LIT MAR record is mostly controlled by the sedimentation rate (Fig. S1).*" and: "*The highly variable LIT MAR record might be a combination of variable sedimentation rates and dry bulk density. For instance, the high LIT MAR around 3.4 and 2.8 ka BP coincide with high sedimentation rates but not a strong change in the dry bulk density or lithogenic material content (Fig. S1). However, the high LIT MAR around 1.4 ka BP coincides with slightly increased sedimentation rates and increased dry bulk density.*"

[Figure]

**Figure S1: a) Content of lithogenic material, b) sedimentation rates, c) dry bulk density and lithogenic mass accumulation rates (LIT MAR) of SO90-63KA. Red stars indicate** [14]**C dates of planktonic foraminifers (Staubwasser et al., 2002, 2003).**

RC2-4: Line 143: the authors state that the last 3 ka show more variable SST that before. Couldn't this be - at least partially - attributed to the fact that the temporal resolution is also higher over the top 3 ka than in some older parts of the record ? I counted, for instance, about 20 data points in the SST record over the 4.5-6.5 ka interval, but ~ 40 points over the 0-2 ka time interval, thus twice as much. Change in sampling resolution should have an effect on our capacity to resolve rapid SST variations. The d18O record, which displays a more constant sampling resolution along the core than the SST record, suggests a relatively similar/invariant type of periodicity across the entire Holocene. Was this record run through Wavelet Analysis? How this compares with the LIT MAR spectral evolution?

Response: We agree with referee #2 that more variable SSTs over the last 3 ka BP than before could be due to the higher temporal resolution in that part of the core. For that reason we have deleted this statement. We have also performed the wavelet analysis for the SST record, but did not see a significant change in the signal.

RC2-5: Wavelet analysis: I wonder why wavelet analyses were not performed/shown for the other proxies as well. Is LIT MAR the only proxy that reveals a shift in periodicity across the Holocene? Shouldn't we expect to see also a change in periodicity in the EM3 record, for instance? Is it seen when performing wavelet analysis?

Response: We performed the wavelet analysis also for other proxies but have focused on the LIT MAR as it shows the clearest signal. For instance, the sedimentation rate show periodicities of about 512 years between ca. 3.5 ka and 2 ka BP (see figure panel a) below) as LIT MAR. However, in contrast to LIT MAR (see figure 3 in manuscript), the periodicities of 1024 to 2048 years are not significant (on a 95% significant level) until 3.5 ka BP but only ca. 6.2 ka BP for the sedimentation rate. The EM3 (see figure panel b) below) show significant (on a 95% significant level) periodicities of about 1024 to 2028 between about 6 and 2 ka BP.

We did not show all wavelet analyses, as we want to focus on the most, in our opinion, important one and do not want to "overload" the manuscript to keep focus on our main statement.

[Figure]

*Wavelet power spectrum for the a) sedimentation rate and b) EM3 of SO90-63KA using a Matlab Wavelet script (Torrence and Compo, 1998). The black line indicates the cone of influence, the dashed black line the 95% significance level.*

RC2-6: Figure 2: Is the "factor 1 – SST" (figure 2b; Giosan et al.) used and discussed in the text ?? I couldn't find a call to this figure 2b in the text.

Response: We have cited Giosan et al. 2018 several times in the earlier version of the manuscript, but did not explicit mentioned that record. However, we agree with referee #2 that there should be a reference to Figure 2b in the discussed text as we show the record in that figure. Therefore, we have added "*Our Holocene alkenone based SST record shows a similar pattern as the nearby planktic foraminifera DNA based SST/winter monsoon record of core Indus 11C near the Indus River mouth (Giosan et al., 2018) (Figure 2b).*" to the manuscript.

---

## Author Response (AR3)

Comments to the Author:
Dear Authors,
thanks for the revisions of your manuscript. You responded well to the last critics. I therefore accept it subject to technical corrections. I see some typos such as a 'S' missing at 'found' line 40. Also % should be spelled out (Line 41). There were only 2 reviewers (Line 433). Please correct those few points.
Best,
Luc Beaufort

Response:
Dear Luc Beaufort, we thank you for accepting our manuscript subject to technical corrections. We went through the manuscript and have made some technical corrections. These include:
- Adding missing spaces and comma (Lines: 18, 178, 317, 402)
- Deleting redundant spaces and words (Lines: 78, 192, 281, 306, 363, 387)
- Spelled out numbers, % and other abbreviations (Lines: 41, 82, 86 – 88)
- Adding missing words and letters (Lines: 61, 91, 109, 123, 130, 137, 181, 204, 223, 245, 249, 256, 335, 345, 351, 362, 397, 419)
- Corrected grammar (Lines: 91, 148, 163, 186, 187, 213, 214, 254, 260, 319, 359, 372, 374)
- Corrected the number of anonymous reviewer (Line 429)

On behalf of all co-authors,
Nicole Burdanowitz